# Sirtuin 1 Regulates Mitochondrial Biogenesis and Provides an Endogenous Neuroprotective Mechanism Against Seizure-Induced Neuronal Cell Death in the Hippocampus Following Status Epilepticus

**DOI:** 10.3390/ijms20143588

**Published:** 2019-07-23

**Authors:** Yao-Chung Chuang, Shang-Der Chen, Shuo-Bin Jou, Tsu-Kung Lin, Shu-Fang Chen, Nai-Ching Chen, Chung-Yao Hsu

**Affiliations:** 1Department of Neurology, Kaohsiung Chang Gung Memorial Hospital, Kaohsiung City 83301, Taiwan; 2Institute for Translation Research in Biomedicine, Kaohsiung Chang Gung Memorial Hospital, Kaohsiung City 83301, Taiwan; 3College of Medicine, Chang Gung University, Taoyuan City 33302, Taiwan; 4Department of Neurology, Kaohsiung Medical University Hospital and School of Medicine, College of Medicine, Kaohsiung Medical University, Kaohsiung City 80708, Taiwan; 5Department of Biological Science, National Sun Yat-Sen University, Kaohsiung City 80424, Taiwan; 6Department of Neurology, Mackay Memorial Hospital and Mackay Medical College, Taipei 10449, Taiwan

**Keywords:** SIRT1, PGC-1α, mitochondrial biogenesis, status epilepticus, hippocampus

## Abstract

Status epilepticus may decrease mitochondrial biogenesis, resulting in neuronal cell death occurring in the hippocampus. Sirtuin 1 (SIRT1) functionally interacts with peroxisome proliferator-activated receptors and γ coactivator 1α (PGC-1α), which play a crucial role in the regulation of mitochondrial biogenesis. In Sprague-Dawley rats, kainic acid was microinjected unilaterally into the hippocampal CA3 subfield to induce bilateral seizure activity. SIRT1, PGC-1α, and other key proteins involving mitochondrial biogenesis and the amount of mitochondrial DNA were investigated. SIRT1 antisense oligodeoxynucleotide was used to evaluate the relationship between SIRT1 and mitochondrial biogenesis, as well as the mitochondrial function, oxidative stress, and neuronal cell survival. Increased SIRT1, PGC-1α, and mitochondrial biogenesis machinery were found in the hippocampus following experimental status epilepticus. Downregulation of SIRT1 decreased PGC-1α expression and mitochondrial biogenesis machinery, increased Complex I dysfunction, augmented the level of oxidized proteins, raised activated caspase-3 expression, and promoted neuronal cell damage in the hippocampus. The results suggest that the SIRT1 signaling pathway may play a pivotal role in mitochondrial biogenesis, and could be considered an endogenous neuroprotective mechanism counteracting seizure-induced neuronal cell damage following status epilepticus.

## 1. Introduction

Epilepsy is a common neurological disorder that may affect more than 70 million people worldwide, though the prevalence differs in developed countries and developing countries, and also varies amongst different socio-economic status in developing countries [1]. Sustained or prolonged epileptic seizures (status epilepticus) are life-threatening neurological emergency situations that frequently lead to the significant damage of cortical neurons, particularly in the hippocampus [2]. Growing evidence has shown that status epilepticus could cause a series of changes at the cellular and molecular levels in the hippocampus, based on both clinical and experimental studies [3,4,5]. These include increased oxidative stress, cytokine activation, and alteration of neuroplasticity, escalating the risk of succeeding epileptic episodes and activating certain late cell death pathways, which may play a crucial role in brain damage, cognitive function decline, and complications with a high mortality rate and high health care costs [2,6]. Prompt management in order to prevent seizure-induced neuronal damage in the brain is an essential goal for the management of status epilepticus [7]. However, the underlying mechanisms of how status epilepticus causes neuronal cell injury or death in the hippocampus still needs to be clarified.

Both animal and clinical studies [8,9,10,11,12,13] have revealed that continuous epileptic seizures can cause the dysfunction of mitochondria, which may act as a critical player in seizure-induced brain damage. It is well understood that prolonged seizures selectively affect Complex I enzyme activity in the mitochondrial respiratory chain, causing excessive oxidative and nitrosative stress, and resulting in neuronal cell injury or death in the hippocampus—which may induce succeeding epileptogenesis [5,12,14]. Consequently, the mitochondria may possess characteristics to determine the fate of neuronal survival after status epilepticus and play a role in therapeutic modification [5,12]. 

Recently, we have demonstrated that prolonged seizures may lead to the damage of the mitochondrial ultra-structure in the hippocampus and the dysfunction of enzyme activity in Complex I of the mitochondrial respiratory chain [10,11,15]. We further validated in our animal model of experimental status epilepticus that a decline in mitochondrial Complex I enzyme activity raised oxidative and nitrosative stress, and increased cytochrome *c* release from the mitochondria to the cytosol—thus triggering the activation of mitochondrial caspase cascades and leading to apoptotic cell death in the hippocampus [10,11,16]. In recent years, mitochondrial dynamics has been acknowledged as a crucial process in affecting cell death and survival. In particular, mitochondrial fission happens as an early event in the apoptotic process and results in neuronal cell death in various cerebral insults [17]. Seizure-affected mitochondrial fission expression with neuronal damage and alteration of mitochondrial dynamic protein expression can provide a protective effect opposing seizure-induced hippocampal neuronal damage [15,18]. Therefore, mitochondrial biogenesis can be considered as a target for potential neuroprotective strategies in status epilepticus and chronic epilepsy.

Sirtuin 1 (SIRT1), a NAD-dependent deacetylase, is a member of the sirtuin family, which acts as an intracellular regulatory protein involving energy homeostasis, development, cell survival, and lifespan [19]. Several important features of SIRT1 have been demonstrated in brain neuronal cells [20]. These include the ability to preserve mitochondria function and modulate responses to DNA damage [19], as well as functionally interact with peroxisome proliferator-activated receptors γ (PPARγ) coactivator 1α (PGC-1α), and may have an important role in mitochondria biogenesis [21]. However, limited studies concerning the SIRT1 pathway in epilepsy and seizure-induced brain injury have been reported. A recent study showed that SIRT1 expression and activity were enhanced in the rat hippocampus following status epilepticus and augmented the PGC-1α/mitochondrial antioxidant signaling pathway [22]. It has been previously reported that the ability of mitochondrial biogenesis is severely impaired in the hippocampus in rats with chronic seizures [15,23]. As mitochondrial biogenesis is an important feature of the SIRT1 pathway, it is tempting to postulate that this pathway could be affected during status epilepticus and may confer protective effects against seizure-induced neuronal damage through the change of mitochondrial biogenesis machinery expression in the hippocampus. This hypothesis was validated using an experimental status epilepticus model in the present study.

## 2. Results

### 2.1. Temporal Changes of SIRT1 and PGC-1α Expression in the Hippocampal CA3 Region Following Status Epilepticus

A previous study showed that SIRT1 activation could enhance the PGC-1α/mitochondrial antioxidant system pathway in pilocarpine-induced status epilepticus [22]. Our aim was to investigate the potential role of mitochondrial biogenesis in the SIRT1/PGC-1α pathway using our previously reported kainic acid (KA)-induced status epilepticus model [10,11]. We first examined whether SIRT1 expression in the hippocampal CA3 subfield was induced following experimental status epilepticus. After unilateral microinjection of KA into the left CA3 region, real-time polymerase chain reaction (PCR) analysis revealed that *Sirt1* mRNA underwent a significant increase in the right hippocampal CA3 area 1 h after the elicitation of sustained hippocampal seizure discharges, followed by a progressive decrement that reached baseline at 24 h with a peak at 3 h (Figure 1A). Western blot analysis of the total protein prepared from the right hippocampal CA3 subfield revealed a significant increase in the expression of SIRT1 from 1 to 48 h with a peak level at 3 h after KA-induced experimental status epilepticus (Figure 1C). Subsequent experiments revealed that PGC-1α expression from the same preparation of the hippocampal CA3 subfield after the microinjection of KA showed a significant increase of *Pgc-1α* mRNA from 1 to 6 h with a peak at 6 h (Figure 1B). The PGC-1α protein expression underwent a significant increase from 3 to 48 h in the hippocampal CA3 subfields with a peak level at 6 h after KA treatment (Figure 1D) that slightly delayed the peaking expression in comparison with SIRT1 expression.

### 2.2. Temporal Changes of Mitochondrial Biogenesis Machinery Expression in the Hippocampal CA3 Region Following Status Epilepticus

PGC-1α is known to regulate mitochondrial biogenesis from which new mitochondria are produced from existing mitochondria [21]. This mitochondrial biogenesis machinery includes the activation of nuclear respiratory factor (NRF) 1 and NRF2, and subsequently mitochondrial transcription factor A (Tfam) [21,24]. The PGC-1α/NRF/Tfam pathway leads to the synthesis of mitochondrial DNA and proteins to generate new mitochondria. To demonstrate the temporal change of mitochondrial biogenesis machinery expression following prolonged seizures, we showed NRF1 expression in the total protein prepared from the right hippocampal CA3 subfield, which revealed a significant increase in the expression of SIRT1 from 3 to 24 h with a peaking level at 6 h after KA-induced experimental status epilepticus (Figure 2A). We further extracted nuclear proteins from the hippocampal CA3 subfield to show the authentic activity of NRF1 as a transcription factor and revealed the increasing DNA-binding activity between 6–24 h after the KA induction of seizures (Figure 2B). 

The mitochondrial DNA transcription factor, Tfam, acts on the promoters and regulates the replication and transcription of the mitochondrial genome [24]. We therefore used the mitochondrial protein fraction to perform Western blot analysis and showed the temporal change of Tfam expression, as well as revealed an increased expression in 3–24 h under KA treatment (Figure 2C). We further showed the extent of mitochondrial biogenesis and determined if it was compatible with NRF1/Tfam expression. Western blot analysis revealed a significant increase of the mitochondrial DNA-encoded polypeptide cytochrome *c* oxidase 1 (COX1) in the hippocampal CA3 subfield in 3–48 h after KA treatment (Figure 2D). Furthermore, we performed the long PCR method to quantify mitochondrial DNA that can be used as an index of mitochondrial DNA (mtDNA) with sufficient integrity [25,26]. The result showed an increased mtDNA content in 3–24 h in the right hippocampal CA3 subfield after KA treatment (Figure 2E). The mitochondrial biogenesis machinery expression following status epilepticus may present endogenous protective effects to counteract the detrimental effect from sustained epileptic activity.

### 2.3. Effect of Sirt1 Knock-Down by Antisense Strategy on PGC-1α Expressions in the Hippocampal CA3 Region Following Status Epilepticus

A temporal correlation was presented between the increased SIRT1 (Figure 1A,C) and upregulated PGC-1α (Figure 1B,D) in the hippocampal CA3 subfield after KA-induced status epilepticus. In order to determine the correlation concerning SIRT1 and PGC-1α in this experimental paradigm, Western blot analysis was applied to test the specificity of an antisense phosphorthioate-modified oligodeoxynucleotide (ODN) for the *Sirt1* gene in the hippocampal CA3 subfield. For conforming the specific effect on SIRT1 expression, after a pre-treated microinjection with antisense ODN for *Sirt1* (1 nmol) into the CA3 subfield, real-time PCR showed a significant decrease in the *Sirt1* mRNA level (Figure 3A), and Western blot analysis revealed a reduction in the SIRT1 protein level in the hippocampal CA3 subfield 3 h after experimental status epilepticus (Figure 3B) when compared with the control groups of sense ODN or scrambled ODN. Corresponding to the change in SIRT1 expression after the pre-treatment of antisense ODN against *Sirt1*, both the *Pgc-1α* mRNA level (Figure 3C) and PGC-1α protein (Figure 3D) expression reduced in 6 h after experimental status epilepticus. Viewed under a laser scanning confocal microscope, immunoreactivity for PGC-1α was low in the hippocampal CA3b neurons in the sham-control animals (Figure 3E-a,b,c) that were immunoreactive to the neuronal marker NeuN. However, many PGC-1α-positive neurons were detected in the hippocampal CA3b area 6 h after experimental status epilepticus in animals with pre-treatment of *Sirt1* sense (Figure 3E-g,h,i) or *Sirt1* scrambled (Figure 3E-j,k,l) ODN. Otherwise, PGC-1α-positive cell was apparently reduced in the hippocampal CA3 neurons 6 h after KA-induced status epilepticus in animals with pre-treated microinjection of *Sirt1* antisense ODN (1 nmoL) into the bilateral hippocampal CA3 field (Figure 3E-d,e,f).

### 2.4. Effect of Sirt1 Knock-Down by Antisense Strategy on Mitochondrial Biogenesis Machinery Expression in the Hippocampal CA3 Region Following Status Epilepticus

We further conducted Western blot analysis to evaluate the effects of *Sirt1* antisense ODN on mitochondrial biogenesis machinery expression in the hippocampal CA3 neurons. After pre-treatment with *Sirt1* antisense ODN (1 nmol) into the bilateral CA3 subfields 24 h before KA-induced status epilepticus, Western blot analysis revealed a reduction in the NRF1 protein level in hippocampal CA3 neurons 6 h after KA treatment when compared with sense and scrambled ODN for *Sirt1* (Figure 4A). With a pre-treatment of *Sirt1* antisense ODN (1 nmoL) into the bilateral hippocampal CA3 area, the DNA-binding activity of NRF1, measured by electrophoretic mobility shift assay (EMSA), also decreased at 6 h after KA-induced sustained seizures (Figure 4B). Viewed under a laser scanning confocal microscope, few immunofluorescence signals of NRF1 (Figure 4C-a,b,c) were expressed in the CA3 neurons that were immunoreactive to the neuronal marker, NeuN, in the sham-control animals. Otherwise, increased immunoreactivity for NRF1 was present in the nucleus of CA3 neurons that were immunoreactive to NeuN at 6 h after the microinjection of KA into the hippocampus in animals that received a pre-treatment of sense (Figure 4C-g,h,i) or scrambled (Figure 4C-j,k,l) ODN for *Sirt1*. Moreover, a decrease in the augmented immunoreactivity for NRF1 detected at 6 h after experimental status epilepticus was found in the hippocampal CA3 neurons (Figure 4C-d,e,f) in animals that were pre-treated with *Sirt1* antisense ODN (1 nmoL) into the bilateral hippocampal CA3 area. We then demonstrated that pre-treatment of *Sirt1* antisense ODN lessened Tfam expression in the hippocampal CA3 subfield 6 h after KA treatment when compared with the sense and scrambled ODN for *Sirt1* (Figure 4D). These effects affect mitochondrial biogenesis, which showed a decreased content of mitochondrial protein COX1 (Figure 4E) and mtDNA (Figure 4F) in rats that received pre-treatment of Sirt1 antisense ODN, and 24 h after induction of experimental status epilepticus when compared with the sham-control, sense ODN, and scrambled ODN groups. These results may reveal the potential correlations between SIRT1 signaling and mitochondrial biogenesis.

### 2.5. Effect of SIRT1/PGC-1α Signaling Pathway on Mitochondrial Respiratory Chain Function, Protein Oxidation, and Neuronal Cell Survival in the Hippocampal CA3 Region Following Status Epilepticus 

We have previously reported [11,15,27] that depression of mitochondrial Complex I enzyme activity in the hippocampus takes place in experimental status epilepticus. We further investigated the effects of *Sirt1* antisense ODN over KA-induced mitochondrial dysfunction, oxidative stress, apoptosis, and neuronal cell survival in the hippocampus. In this series of experiments, we showed the depressed mitochondrial Complex I activity with pre-treated *Sirt1* antisense ODN (Figure 5A) three days after induction of experimental status epilepticus when compared with sham-control, sense ODN, and scrambled ODN groups. It also revealed increased protein oxidation (Figure 5B) and an augmented extent of activated caspase-3 expression in the hippocampal CA3 subfield seven days after the induction of experimental status epilepticus (Figure 5C). Increased extent of neuronal damage with *Sirt1* antisense ODN treatment were demonstrated in the hippocampal CA3 subfield in both qualitative (Figure 5D) and quantitative (Figure 5E) analysis of DNA fragmentation, an index for apoptosis, seven days after the induction of status epilepticus when compared with the sham-control, sense ODN, and scrambled ODN groups. These results may imply that SIRT1/PGC-1α signaling involves neuronal survival in status epilepticus.

## 3. Discussion

Based on an experimental animal model that clinically connected in status epilepticus, the present study demonstrated the changes in mitochondrial biogenesis machinery and neuronal cell survival in the hippocampus following status epilepticus. Particularly, we noted that SIRT1 expression was enhanced in the rat hippocampus following status epilepticus and upregulated PGC-1α expression. Downregulation of SIRT1 reduced PGC-1α expression and impaired the mitochondrial biogenesis. *Sirt1* gene knock-down by the antisense strategy, also accompanied by mitochondrial respiratory chain dysfunction, increased oxidative stress, further heightened caspase-3 activity, and augmented neuronal damage in the hippocampal CA3 subfield. These results may indicate that the SIRT1 pathway involves mitochondrial biogenesis machinery and exerts an endogenous neuroprotective mechanism during status epileptics. 

SIRT1, the most-studied sirtuin family, was initially discovered to deacetylate histones, but was later also shown to deacetylate other target proteins such as PGC-1α, HIF1α, p53, FoxO1, and Notch [19,28]. It plays a critical role in many physiological activities, such as the regulation of gene transcription, DNA damage repair, protein secretion, metabolic action, cell differentiation, and cell survival [29]. These characteristics hold potential as therapeutic targets for various human disorders, including malignancy, metabolic disorders, advanced heart failure, and neurodegenerative diseases [29,30]. SIRT1 is distributed in various adult brain areas, with higher expression in the neuronal cells of the cortex, hippocampus, cerebellum, and hypothalamus, and lower expression in white matter [31]. Evidence has revealed that an enhanced SIRT1 and PGC-1α signaling pathway may counteract reactive oxidative species (ROS), promote mitochondrial biogenesis, and enhance mitochondrial functions in the neuronal cells, decrease neuronal and glial cell inflammation, and involve the pathway of neuron cell death and survival [29,30,32,33,34]. SIRT1 has been described as cleavage at the nuclear full-length SIRT1 (110 kDa) to generate a stable but enzymatically inactive 75 kDa fragment of SIRT1 under high levels of ROS or proinflammatory cytokines during injury and promote chondrocyte survival [35,36]. Full-length FLSirT1 possesses the capacity to modulate mitochondrial levels of Bax and Bcl2 in apoptosis [37]. In the situation of status epilepticus, we propose that the nuclear full-length SIRT1, apart from deacetylating PGC-1α, is a protective mechanism of SIRT1 via mitochondrial biogenesis. Recent studies have demonstrated that the SIRT1/PGC-1α signaling pathway exerts potential neuroprotective properties in neurodegenerative diseases [38,39] such as Alzheimer’s disease [40], Parkinson’s disease 41], Huntington’s disease [39], acute stroke [41,42], chronic epilepsy, and status epilepticus [43,44,45,46]. Therefore, the SIRT1 can be considered as a target for potential neuroprotective strategies in epilepsy. 

Prolonged epileptic discharges can trigger a series of events with enormous changes at the cellular and molecular levels of neurons. It is well known that mitochondria, which produce the majority of cellular energy, have been shown to play a crucial role in the mechanisms of cell death in epilepsy [4,5,47]. While the oxidative phosphorylation of mitochondria offers the main source of ATP in neurons, is involved in cellular Ca(2+) homeostasis; however, in the meantime, it generates reactive oxidative species (ROS) as a byproduct, causing dysfunction of mitochondria as in condition with excessive ROS formation and may greatly affect the excitability and synaptic transmission of neurons that are crucial in the pathophysiology of epilepsy [48]. In earlier studies of mitochondria, the focus was mainly on the bioenergetics role; however, in recent years, with the development in animal models, molecular biology, imaging techniques, and systems-based approaches, the scope of mitochondrial research has been rapidly changing. We have witnessed the significant advancement of mitochondrial research in a wide variety of cellular functions and signaling events, such as the apoptotic process, mitochondrial biogenesis, mitochondrial dynamics, mitophagy, and the role of immunity among various neurological disorders [49,50].

Neuroprotection with status epilepticus should encompass not only the prevention of neuronal cell death, but also the preservation of neuronal and network function [51]. Except for the detrimental consequences following status epilepticus, acute response protein to counteract these detrimental effects may be elicited as an endogenous protective mechanism. Endogenous mechanisms for neuronal cell survival following prolonged seizure insult are those that have been evolutionarily conserved and may trigger a number of signaling pathways to exert a protective effect, and are therefore strong candidates to apply as therapeutic strategies [51]. In animal studies, several endogenous protective mechanisms to reduce seizure-induced neuronal damage following status epilepticus have been proposed, including epileptic tolerance, enhanced vascular endothelial growth factors, activation of the ERK1/2 pathway, and activation of the adenosine A1 receptors, erythropoietin receptor, and mitochondrial uncoupling protein 2 (UCP2) [27,51,52,53,54,55]. In the present study, we noted that the *Sirt1* mRNA underwent a significant increase after KA-induced status epilepticus, followed by an increase in the SIRT1 protein level. Thus, our results suggest that SIRT1 may play an endogenous neuroprotective role against hippocampal neuronal cell damage under the excitotoxic stress of sustained epileptic seizures.

In our recent works, we showed that sustained seizures prompted the overproduction of nitric oxide, superoxide anion, and peroxynitrite formation that compromised mitochondrial respiratory chain enzyme activity, particularly in Complex I, and mitochondrial ultrastructure damage that caused cytochrome c/caspase-3-dependent apoptotic cell death in the hippocampal CA3 subfield following experimental status epilepticus [10,11,16]. With this status epilepticus model, we demonstrated that activation of PPARγ increased mitochondrial UCP2 and superoxide dismutase 2, decreased mitochondrial translocation of Bax, reduced cytosolic release of cytochrome *c* through stabilizing the mitochondrial transmembrane potential, and lessened apoptotic neuronal cell death in the hippocampus [27]. Furthermore, we revealed that activation of the phosphorylation of dynamin-related protein 1 (Drp1), the major mitochondrial fission protein, was related to seizure-induced neuronal damage and that lessened *p*-Drp1 expression could reduce mitochondrial fission, as well as decrease mitochondrial dysfunction and oxidation, which offer a protective strategy against seizure-induced hippocampal neuronal damage in the hippocampus following status epilepticus [15]. Recently, we found that inhibition of mitochondrial respiratory chain Complex I enzyme activity by rotenone may change ionic currents and miniature end-plate potential in mouse hippocampal neurons [14]. These events suggest that the mitochondrial biogenesis and bioenergetic function of mitochondria may be related to hippocampal neuronal cell damage and further epileptogenesis under the conditions of epilepsy or status epilepticus.

SIRT1 possesses anti-aging effects as well as coordinating metabolic demands [19,34], and is known to functionally interact with PGC-1α [21]. SIRT1 in the brain activates the central pacemaker to maintain robust circadian control in young animals, and a decay in this activity may play an important role in aging [56]. Whereas the levels of SIRT1 may decrease in situations such as aging or inflammation [56,57], evidence has demonstrated that the SIRT1 signaling pathway is activated in various neurological disorders [38,39,40,41,42,43,44,45,46,58]. These diverse results may be related to the different effects of physiological conditions or pathological injuries. Given the diverse role of both SIRT1 and PGC-1α in regulating cellular metabolism and homeostasis, this is known to exert protective effects in various acute and chronic neurological diseases, such as cerebral ischemia and neurodegenerative diseases [34,39,42,59,60,61]. Limited studies have explored the mechanism of the SIRT1 pathway in chronic epilepsy or status epilepticus [43,62]. SIRT1 has been noted to be upregulated in patients with epilepsy and in rat models following acute seizure or status epilepticus [22,63]. A recent report showed that SIRT1 expression and activity were enhanced in the rat hippocampus following status epilepticus, and augmented the PGC-1α/mitochondrial antioxidant signaling pathway [22]. As mitochondrial biogenesis is another important feature of the SIRT1/PGC-1α signaling pathway [59,64,65], in the present report, we demonstrated that SIRT1 regulates the mitochondrial biogenesis machinery and affects the hippocampal neuronal cell fate following KA-induced status epilepticus. We first confirmed that *Sirt1* and *Pgc-1α* mRNA levels were elevated in a quick fashion from 1 to 6 h, and SIRT1 and PGC-1α expressions were also quickly activated from 1 h and lasted at least 48 h with, peaking levels at 3 h (SIRT1) and 6 h (PGC-1α) after KA induction of status epilepticus in the hippocampus. The mechanism of activation of SIRT1 and PGC-1α during epilepsy or status epilepticus is unclear. However, these phenomena are seen in various brain insults, such as acute brain injury, hemorrhage, ischemia, hypoxia, and even in the model of neurodegenerative diseases [34,59,61,66,67]. We suggest that in the brain neuronal cells, the stress and excitotoxicity related to prolonged seizures may activate endogenous SIRT1 expression and trigger PGC-1α and their downstream pathways, including mitochondrial biogenesis. Thus, this may denote that SIRT1 may possess an endogenous neuroprotective mechanism in the nervous tissue of living organisms.

Mitochondrial biogenesis machinery represents a complex biological process that controls the biogenesis of mitochondria and the maintenance of mtDNA. Most respiratory proteins and all of the proteins and enzymes involved in mtDNA replication, transcription, and translation, as well as gene products necessary for the numerous mitochondrial functions stem from nuclear genes [68,69,70]. Nuclear respiratory factors, NRF1 and NRF2, which function as transcriptional regulators, are essential subunits of the oxidative phosphorylation system and also regulate the expression of many other genes involved in mtDNA replication [24,68,69]. Tfam is a transcription factor that acts on the promoters within the D-loop region of mtDNA and regulates the replication and transcription of the mitochondrial genome [24,68]. The TFAM gene contains consensus-binding sites for both NRF1 and NRF2, which offers an exclusive mechanism for the living cell to integrate nuclear DNA-encoded proteins with the transcriptional factor for mtDNA generation [24,68]. Previous studies have shown that PGC-1α binds to and co-activates the transcriptional function of NRF1 on the promoter for Tfam [30,71]. In the current study, we demonstrated the activation of mitochondrial biogenesis machinery in the hippocampal CA3 subfield after experimental status epilepticus. Both the NRF1 protein expression and NRF1 DNA-binding activity increased after prolonged seizures. In accordance with the NRF1 finding, Tfam, the regulator of the mitochondrial genome, showed a similar fashion of expression in the hippocampal CA3 subfield. As such, the increased mitochondrial-encoded gene COX1 may indicate the activation of mitochondrial biogenesis. As previously reported [25,26], the long PCR method quantifies intact mtDNA and can be used a reliable indicator for mitochondrial biogenesis. We then showed the temporal changes of mtDNA content in the hippocampal neurons following experimental status epilepticus. These results strengthen the thought that prolonged epilepsy activates mitochondrial biogenesis machinery in the hippocampal CA3 subfield and these changes may have a crucial biological role in status epilepticus. 

Although the SIRT1 pathway is known to regulate PGC-1α/mitochondrial biogenesis and has been reported in various neurological conditions [34,59,60,61], at present, there is no report available on seizure disorders including status epilepticus. In the present study, while we could not find a significant change in the electrophysiology during the control SIRT1 pathway, we demonstrated that downregulation of SIRT1 decreased PGC-1α/mitochondrial biogenesis machinery expression including NRF1 protein expression, DNA-binding activity, and the expression of Tfam, a transcription factor vital for regulating the replication and transcription of the mitochondrial genome [24]. Downregulation of SIRT1 also lessened the amount of mtDNA as well as mitochondrial DNA encoded COX1 expression, which was accompanied by impaired mitochondrial respiratory function, increased protein oxidation, raised apoptotic process, and neuronal damage. As PGC-1α is a key player in mitochondrial biogenesis [72,73], and downregulation of SIRT1 also decreased PGC-1α expression, it is tempting to speculate on the intimate relationship between SIRT1/PGC1α and mitochondrial biogenesis in prolonged seizures. The ability to generate more mitochondria may reflect the capability of our bodies to cope with various insults, such as cerebral ischemia, seizure disorder, or neurodegenerative diseases. Another crucial feature of the SIRT1 pathway is the ability to regulate the antioxidant system in our bodies to counteract the detrimental effect of excessive oxidative stress from numerous clinical conditions. Currently, the relative importance between mitochondrial biogenesis and the antioxidant system concerning the beneficial effects of the SIRT1 pathway awaits further exploration and clarification. In Figure 6, we illustrated the proposed SIRT1 signaling pathway involved in PGC1α/mitochondrial biogenesis that exerted an endogenous protective mechanism in the hippocampal neuronal tissue following experimental status epilepticus.

## 4. Materials and Methods 

### 4.1. Animals

Experimental procedures in the present study were accomplished in accordance with the guidelines on animal experimentation, endorsed by the constituted research ethics committee of Kaohsiung Chang Gung Memorial Hospital, Taiwan (project identification code 2008061901 was approved by the constituted research ethics committee at Kaohsiung Chang Gung Memorial Hospital, Taiwan; approved on 19 June 2006). Experiments were carried out in 360 specific pathogen-free adult male Sprague-Dawley rats (250–300 g) that were purchased from BioLASCO Taiwan Co. Ltd. (Taipei, Taiwan) and housed in the isolated cages of the Center for Laboratory Animals at Kaohsiung Chang Gung Memorial Hospital with the temperature maintained at 24 ± 1 °C and light:dark at a 12:12 h cycle. Rat chow and tap water were available ad libitum in the laboratory. All efforts were made to decrease the number of experimental animals and to minimize animal suffering during the experiment.

### 4.2. Experimental Status Epilepticus

The model of experimental status epilepticus in rats has been well established in our laboratory [10,11,74]. After anesthesia induced by the inhalation of 3% isoflurane, the head of the rat was fixed to a stereotaxic headholder (Kopf, Tujunga, CA, USA), and the body of the rat was maintained at a temperature of 37 °C by placing on a heating pad. Kainic acid (KA; 0.5 nmoL; Tocris Cookson, Bristol, UK), a potent neuroexcitatory amino acid agonist that acts by activating receptors for glutamate, dissolved in 0.1 M phosphate buffered saline (PBS, pH 7.4) was microinjected into the left side CA3 subfield (reference: 3.3–3.6 mm posterior to bregma, 2.4–2.7 mm from the midline, and 3.4–3.8 mm below the cortical surface) of the hippocampus consistently resulted in concomitant and progressive increases in both the root mean square and mean power frequency values of seizure electroencephalographic (hEEG) activity, which can be recorded from the bilateral hippocampus [10,11]. This animal model clinically simulated temporal lobe status epilepticus. The seizure-like activity of hEEG was ensured to continue for 60 min before termination by diazepam (30 mg/kg) intraperitoneally [11,74]. The animals were then injected with penicillin (10,000 IU; YF Chemical Corporation, Taipei, Taiwan) intramuscularly to prevent post-surgical infections. For recovery, rats were returned to the cages individually. Rats that received anesthesia and surgical preparations without additional experimental manipulation served as sham control groups. 

### 4.3. Gene Knockdown by Microinjection of Oligonucleotides into the Hippocampus

Gene knockdown was conducted with an antisense phosphorthioate-modified oligodeoxynucleotide (ODN) against the *Sirt1* gene: 5′-ATACCATTCTTTGGTCTAGA-3′. The corresponding sense *Sirt1* ODN: 5′-TCTAGACCAAAGAATGGTAT-3′ and scrambled *Sirt1* ODN: 5′-GCTTTTATCATTACGATCG-3′ were used as controls. The ODNs were customer-made by Protech Technology (Taipei, Taiwan). Microinjection of the antisense, sense, or scrambled *Sirt1* ODN (500 pmol in 100 nL) into the bilateral hippocampal CA3 subfield was performed 24 h before the KA-induced experimental status epilepticus. 

### 4.4. Collection of the Hippocampal Tissue

After the experiments, the rats were anesthetized and perfused intracardially with normal saline at time-intervals of 1, 3, 6, 24, or 48 h, or 3, or 7 days [10,15,74]. The brain tissue was dissected and placed on a piece of gauze moistened with ice-cold normal saline [10,15,74]. Routinely, we collected the right side of the hippocampal CA3 tissues samples (recording side of the hEEG) for further experiments [74]. This allowed us to ascertain that the results of the analysis were directly due to sustained seizure activities, instead of KA-induced cell toxicity. The other hippocampal tissue samples were stored at −80 °C until biochemical analyses. 

### 4.5. RNA Isolation and Reverse Transcription Real-Time Polymerase Chain Reaction

For quantitative analysis of *Pgc-1α* mRNA expression in the hippocampal CA3 subfield at 1, 3, 6 h, 24, or 48 h after KA-induced status epilepticus, the total RNA from the hippocampal CA3 subfield was isolated with a RNeasy mini kit (Qiagen, Duesseldorf, Germany) according to the manufacturer’s protocol, as per our previous report [74]. Reverse transcriptase (RT) reaction was performed using an ImProm II^TM^ Reverse Transcription System (Promega, Madison, WI, USA) for the first-strand cDNA synthesis. Real-time polymerase chain reaction (PCR) for the amplification of cDNA was performed using a LightCycler (Roche Diagnostics, Mannheim, Germany). PCR for each sample was carried out in duplicate for all cDNAs and for the glyceraldehyde-3-phosphate dehydrogenase (GAPDH) control [74]. The primer pairs for the amplification of *Pgc-1α* and GAPDH cDNA used in this study were as follows:*Sirt1*:forward: 5’-TCGTGGAGACATTTTTAATCAGG-3’;reverse: 5’-CAGTGTCCGAGTCTGAATCCT-3’.*Pgc-1α*:forward: 5’-GTTTCATTACCTACCGTTACAC-3’; reverse: 5’-ATCGTCTGAGTTTGAATCTAGG-3’.GAPDH:forward: 5′-CAACTCCCATTCTTCCACCT-3′; reverse: 5′-GCCATATTCATTGTCATACCAG-3′.

The PCR products were subsequently subjected to agarose gel electrophoresis for further confirmation of amplification specificity. Fluorescence signals from the amplified products were quantitatively assessed using the LightCycler software program (version 3.5). The second derivative maximum mode was chosen with the baseline adjustment, set in the arithmetic mode. The relative change in *pgc-1α* mRNA expression was determined by the fold-change analysis [74], where fold change = 2^−[ΔΔC*t*]^ and ΔΔC*t* = (C*t,_pgc-1α_* − C*t,*_GAPDH_). Note that the C*t* value is the cycle number at which the fluorescence signal crosses the threshold.

### 4.6. Western Blot Analysis

Western blot analysis was performed on the proteins extracted from the total lysate, or from the nuclear, cytosolic, or mitochondrial fractions of hippocampal CA3 tissues. The primary antisera was used, including a mouse monoclonal or polyclonal antiserum, against the SIRT1 (sc-74465, Santa Cruz Biotechnology, Dallas, TX, USA), Tfam (ab131607, Abcam, Cambridge, UK), or COX1 (ab14705, Abcam); or a rabbit polyclonal antiserum against PGC-1α (sc-13067, Santa Cruz Biotechnology), NRF1 (sc-33771, Santa Cruz Biotechnology), COX IV (ab16056, Abcam), or β-actin (ab8227, Abcam). This was followed by incubation with the secondary antisera, including horseradish peroxidase-conjugated goat anti-mouse IgG (115-035-003, Jackson ImmunoResearch, West Grove, PA, USA) for SIRT1, Tfam, and COX1; or goat anti-rabbit IgG (111-035-045, Jackson ImmunoResearch) for PGC-1α, NRF1, COX IV, and β-actin. The specific antibody–antigen complex was detected by an enhanced chemiluminescence western HRP substrate (Merck Millipore, Billerica, MA, USA). The amount of protein was quantified by ImageJ software (National Institutes of Health, Bethesda, MD, USA) and was expressed as the ratio relative to β-actin protein or COX IV (as the mitochondrial loading control).

### 4.7. Double Immunofluorescence Staining and Laser Confocal Microscopy

According to our previous studies [15,74], double immunofluorescence staining was accomplished by using a goat polyclonal antiserum against PGC-1α and NRF1 (Santa Cruz Biotechnology), and a mouse monoclonal antiserum against neuron-specific nuclear protein (a specific neuronal marker, NeuN; Chemicon). The secondary antisera contained a goat anti-rabbit IgG conjugated with AlexaFluor 488 and a goat anti-mouse IgG conjugated with Alexa Fluor 568 (Molecular Probes, Eugene, OR, USA). Tissue sections were examined under an Olympus AX-51 epifluorescence microscope (Olympus, Kyoto, Japan); immunoreactivity for NeuN exhibited red fluorescence, and PGC-1α and NRF1 manifested green fluorescence.

### 4.8. Electrophoretic Mobility Shift Assay (EMSA)

NRF1 binding activities in nuclear protein from the hippocampal CA3 samples following sustained seizures were measured by the electrophoretic mobility shift assay (EMSA) [25,26,74]. The following oligonucleotides in the binding assays after hybridization were used to obtain the corresponding DNA duplex: NRF1 cons5, 5-TCAGAGGGGCCTGCGGCTAT-3, and NRF1 cons3, 5-ATAGCCGCAGGCCCCTCTGA-3.

According to the manufacturer’s instructions (Roche Molecular Biochemicals, Mannheim, Germany), the oligonucleotide was labeled with the Dig-ddUTP solution. The binding reaction was performed in a mixed solution (20 μL) that contained binding buffer (pH 7.6; 10 mM of Tris-HCl, 20 mM of NaCl, 1 mM of DTT, 1 mM of EDTA, and 5% of glycerol), 1 μg of poly(dI-dC), 0.5 ng of Dig-labeled probe, and 30 μg of nuclear proteins. After 20 min of incubation at room temperature, the mixture was administered to electrophoresis on a 6% nondenaturing polyacrylamide gel at 180 V for 2 h under a low ionic strength circumstance. The gel was then transferred to a positively charged nylon membrane and cross-linked by a UV crosslinker. 100-fold excessive unlabeled oligonucleotides were conducted for EMSA for competitive binding assay. For super-shift assays, the binding reaction was included with the addition of a specific antibody for NRF1 (1 μg/reaction), 1 h prior to the introduction of labeled oligonucleotide probes.

### 4.9. Long Polymerase Chain Reaction (PCR) For Quantitation of Mitochondrial DNA

According to previous reports [25,26], the long polymerase chain reaction (PCR) method produces reliable quantification of entirely intact rat mitochondrial DNA (mtDNA), with the use of mouse mtDNA as an internal control. Briefly, the reactions (10 mL) contained 0.4 ng of total DNA extracted from the rat hippocampus, 4 pmoL of each oligonucleotide primer, 0.5 U of the LA Taq enzyme (Takara Bio., Kusatsu, Japan), and 400 mmoL/L of dNTP combinations. The same amount (0.4 ng) of total DNA obtained from mouse brains serving as an internal standard control was added to the PCR reaction mixtures. The primers to amplify 14.3-kb mitochondrial genomes for both rat and mouse were: forward; 5’-ATATTTATCACTGCTGAGTCCCGTGG-3’ and reverse; 5’-AATTTCGGTTGGGGTGACCTCGGAG-3’. The details for long PCR was according to our previous studies [25,26], and the PCR products were digested with the restriction enzyme NcoI (Promega, Madison, WI, USA) under 37 °C for 2 h and fractionated through 1% agarose gel. Whereas the 14.3-kb fragment amplified from the mtDNA of rat hippocampal neurons, the 7.0- and 7.3-kb restriction fragments as one band, representing the amplified mouse mtDNA, served as an internal control [25,26]. The signal intensities of these bands were assessed by image analysis, followed by quantitative densitometry with ImageJ (National Institutes of Health, Bethesda, MD, USA).

### 4.10. Assays for the Activity of Complex I of Mitochondrial Respiratory Chain Enzyme 

Isolation of mitochondria from the hippocampal CA3 was according to our previous reports [10,15]. The activity of Complex I of the mitochondrial respiratory chain enzyme (nicotinamide adenine dinucleotide (NADH) ubiquinone oxidoreductase) was analyzed based on the enzyme activity immunocapture assays [15]. Ninety-six well plates coated with monoclonal antibodies against the oxidative phosphorylation Complex I (ab109721, Abcam) were performed according to the manufacturer’s recommendations. The Complex I activity was measured by adding an assay solution and the oxidation of NADH was monitored by measuring its decrease in absorbance at 450 nm in kinetic mode at 30 °C for 2 h. Assays for Complex I activity were evaluated using a Multiskan Spectrum reader (Thermo Scientific, Miami, OK, USA). Each tissue sample was measured at least in duplicate.

### 4.11. Measurement of Protein Oxidation (Oxidative Stress)

The extent of oxidized protein was measured using a detection kit (OxyBlot, Chemicon, Temecula, CA, USA). Hippocampal proteins extracted from the CA3 tissues were reacted with 2,4-dinitrophenylhydrazine and derivatized to 2,4-dinitrophenylhydrazone (DNP-hydrazone) [15,74]. The tissue samples of DNP-derivatized proteins were separated on a 15% SDS-polyacrylamide gel, and then introduced to Western blot analysis. The obtained blots were reacted with a rabbit anti-DNP antibody, followed by incubation with a horseradish peroxidase-conjugated goat anti-rabbit IgG according to the manufacturer’s instructions.

### 4.12. Qualitative and Quantitative Analysis of DNA Fragmentation

After total DNA was extracted from the CA3 tissues, nucleosomal DNA ladders were amplified by a PCR kit for DNA ladder assays (Maxim Biotech, San Francisco, CA, USA) to enhance the detection sensitivity, and were separated by electrophoresis on 1% agarose gels [11,74]. To quantify apoptosis-related DNA fragmentation, a cell death ELISA kit (Roche Molecular Biochemicals, Mannheim, Germany) was performed to evaluate the level of histone-associated DNA fragments in the cytoplasm. Proteins extracted from the CA3 tissues served as the source of antigen, together with the primary anti-histone antibody and secondary anti-DNA antibody coupled to peroxidase. The level of nucleosomes in the cytoplasm was quantitatively determined using 2,2′-azino-di-[3-ethylbenzthiazoline] sulfonate as the substrate. Absorbance was measured at 405 nm with a reference at 490 nm using a Multiskan Spectrum reader (Thermo Scientific, Miami, OK, USA).

### 4.13. Statistical Analysis

The continuous variables were expressed as the mean ± standard error of the mean (SEM). One-way analysis of variance (ANOVA), followed by post hoc analysis using the Scheffé multiple range tests, was applied to compare the group mean differences. A *p*-value of <0.05 was considered statistically significant.

## 5. Conclusions

Mitochondrial dysfunctions occur as a consequence of status epilepticus and promote seizure-induced neuronal cell death. Mitochondrial biogenesis and bioenergetics can be considered as a target for potential neuroprotective strategies in epilepsy. We suggest that activation of the SIRT1 signaling pathway may promote PGC-1α expression and the ability of mitochondrial biogenesis, enhance mitochondrial function, and lessen ROS. Therefore, it could be considered as an endogenous neuroprotective mechanism, counteracting seizure-induced neuronal cell damage following status epilepticus. The ability to enhance SIRT1 signaling may provide new insights into the development of more effective neuroprotective strategies against seizure-induced brain damage, and the design of novel management perspectives against multiple drug-resistant forms of epilepsy and status epilepticus.

## Figures and Tables

**Figure 1 ijms-20-03588-f001:**
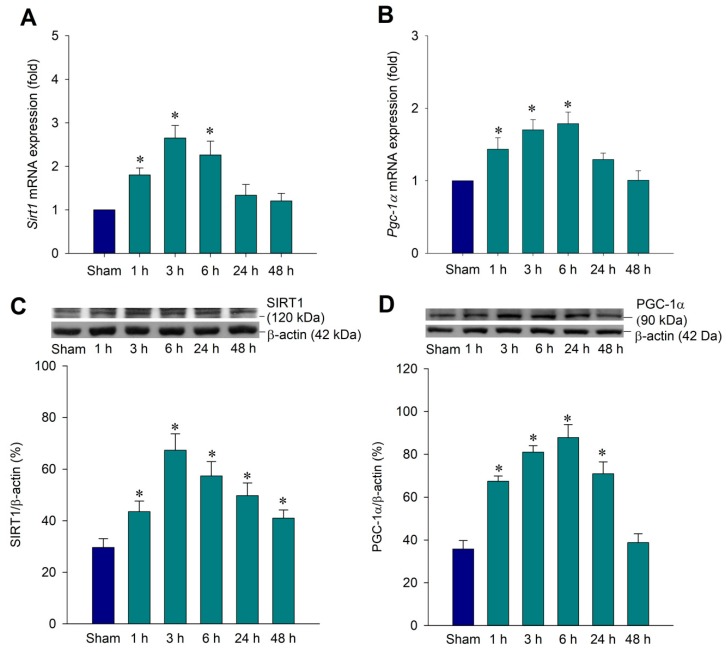
Upregulation of the expression of SIRT1 and PGC-1α mRNA and proteins after the microinjection of kainic acid (KA) in the hippocampal CA3 subfield. Upregulation of the expression of *Sirt1* mRNA (**A**) and *Pgc-1α* mRNA (**B**) detected in samples were collected from the right CA3 subfield of the hippocampus at 1, 3, 6, 24, or 48 h after the microinjection of 0.5 nmol KA into the left hippocampal CA3 subfield or sham animals. Representative gels (inset) or changes in SIRT1 protein (**C**) and PGC-1α protein (**D**), relative to β-actin, detected in samples were collected from the right CA3 subfield of the hippocampus under the same experimental conditions. Values are the mean ± standard error of the mean (SEM) of quadruplicate analyses from six animals per experimental group. * *p* < 0.05 versus the sham-control group in the Scheffé multiple-range test.

**Figure 2 ijms-20-03588-f002:**
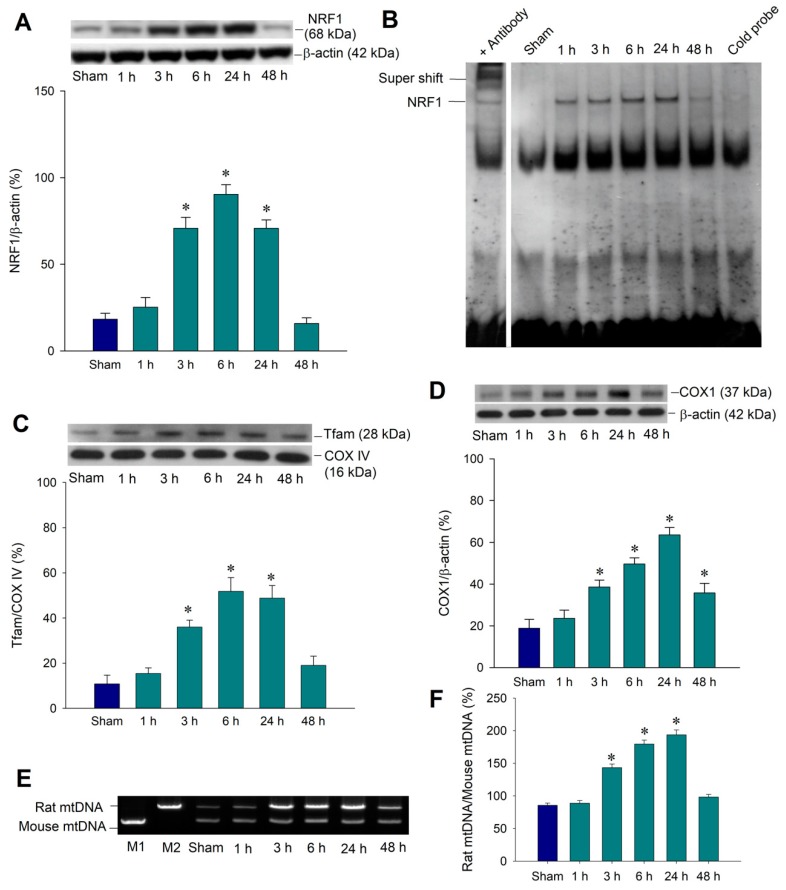
Involvement of mitochondrial biogenesis in kainic acid (KA)-induced experimental status epilepticus in the hippocampal CA3 subfield. (**A**) Temporal changes in the NRF1 protein relative to β-actin protein, and (**B**) representative gel depicting the electrophoresis mobility shift assay of the NRF1 DNA-binding activity in nuclear extracts from the right CA3 subfield of the hippocampus 1–48 h after the microinjection of KA (0.5 nmol) into the left hippocampal CA3 subfield or sham animals. (**C**) Mitochondrial fraction of samples collected 1–48 h after the microinjection of KA (0.5 nmoL) into the left hippocampal CA3 subfield for Tfam expression. COX IV was used as the internal loading control for the mitochondrial fraction. (**D**) Representative gels (inset) or temporal changes in COX I protein relative to β-actin protein were demonstrated. (**E**) Long PCR for the quantitation of mitochondrial DNA (mtDNA) revealed a temporal change in mtDNA in rats 1–48 h after the microinjection of KA (0.5 nmol) into the left hippocampal CA3 subfield or in sham animals. M1: Mouse mtDNA control marker; M2: Rat mtDNA control marker. (**F**) Quantification of mtDNA from rat hippocampal neurons was presented as a rat mtDNA to mouse mtDNA ratio. Values are the mean ± SEM of the ratio of β-actin, or COX IV to the loading controls, and are quadruplicate analyses from six animals per experimental group in (**A**,**C**,**D**,**F**). * *p* < 0.05 versus the sham-control group in the Scheffé multiple-range test.

**Figure 3 ijms-20-03588-f003:**
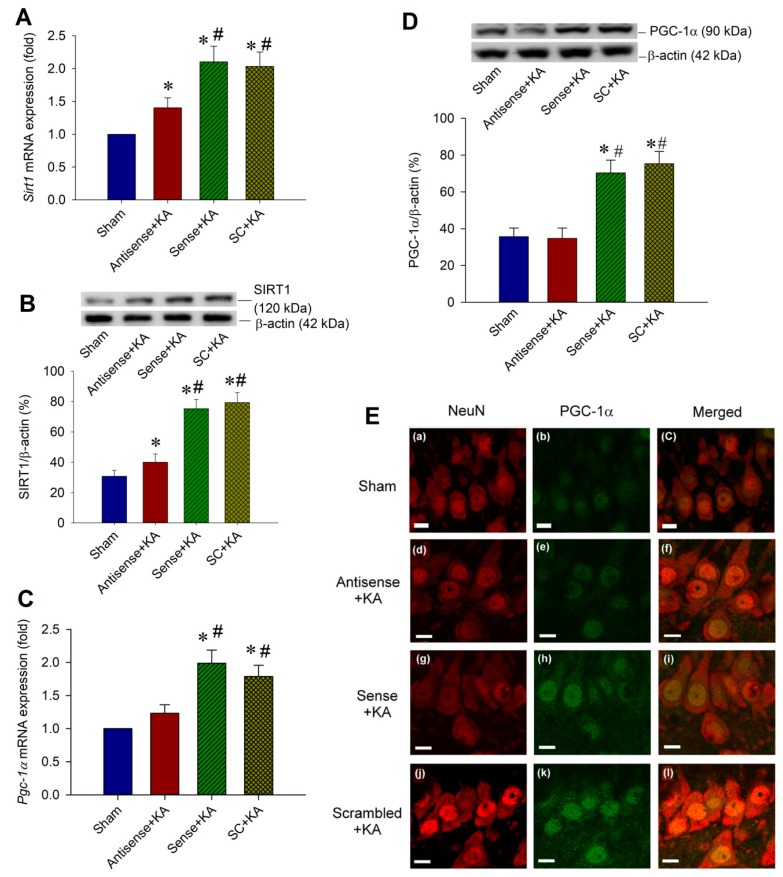
Blockade of PGC-1α expression by antisense oligodeoxynucleotides (ODNs) against *Sirt1*. Rats were microinjected into bilateral CA1 subfields with sense ODN, antisense ODN, or scrambled (SC) ODN for Sirt1, 1 nmol each, 24 h before the microinjection of KA into the left hippocampal CA3. (**A**) Changes of *Sirt1* mRNA expression from the CA3 subfield of the hippocampus 3 h after the microinjection of KA (0.5 nmol) into the left hippocampal CA3 subfield, and received 24 h before pre-treatment with application into the bilateral CA3 subfield of sense, antisense, or SC ODNs for *Sirt1* (1 nmol). (**B**) Representative gels (inset) or changes in SIRT1 relative to β-actin from the CA3 subfield of the hippocampus 3 h after the same experiments. (**C**) Changes of Sirt1 mRNA expression from the CA3 subfield of the hippocampus 6 h after the microinjection of KA (0.5 nmoL) into the left hippocampal CA3 subfield, and received 24 h before pre-treatment with application into the bilateral CA3 subfield of sense, antisense, or SC ODNs for Sirt1 (1 nmol). (**D**) With the same experimental conditions in (**C**), Western blot showed the changes of PGC-1α expression relative to β-actin (42 kDa), 6 h after the microinjection of KA (0.5 nmol) and various ODN against *Sirt1* (1 nmoL). Values are the mean ± SEM of quadruplicate analyses from six animals per experimental group. * *p* < 0.05 versus the sham control group, # *p* < 0.05 versus the *Sirt1* antisense ODN + KA group in the Scheffé multiple-range test. (**E**) Laser scanning confocal microscopic images of a representative cell in the right CA3b subfield of the hippocampus that was immunoreactive to a neuronal marker, NeuN (red fluorescence) and additionally stained for PGC-1α (green fluorescence) in the sham-control animals (**E-a,b,c**), or 6 h after the microinjection of KA (0.5 nmol) into the left hippocampal CA3 subfield in animals that received pre-treatment with application into the bilateral CA3 subfield 24 h after animals received antisense (**E-d,e,f**), sense (**E-g,h,i**), or SC (**E-j,k,l**) ODN against *Sirt1* (1 nmoL). Scale bar, 10 μM.

**Figure 4 ijms-20-03588-f004:**
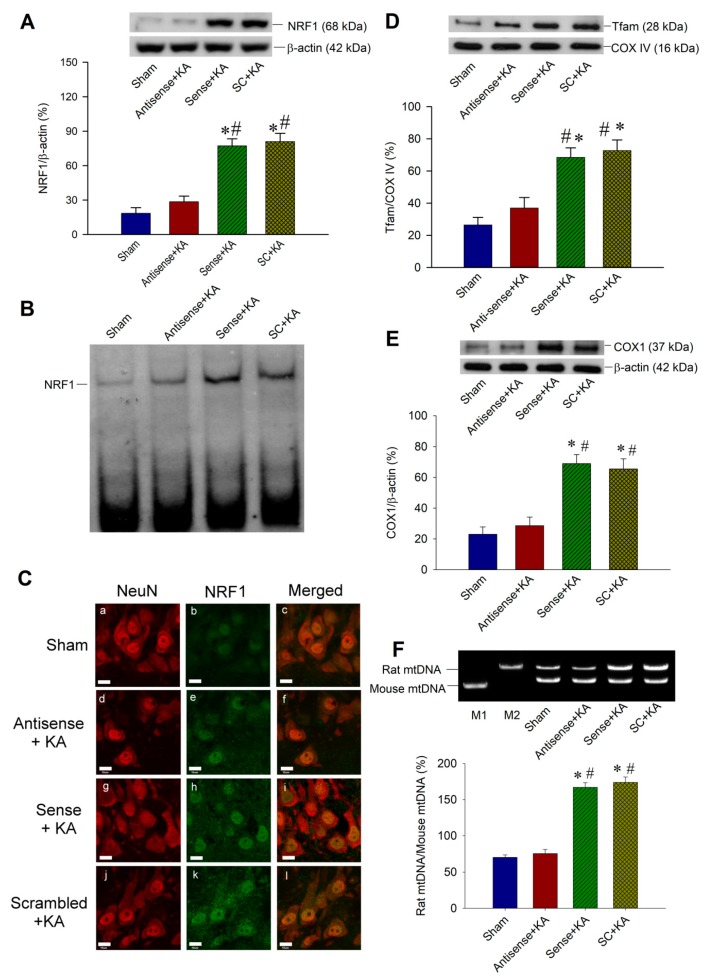
Effects of antisense ODNs against *Sirt1* on mitochondrial biogenesis in kainic acid (KA)-induced status epilepticus in the hippocampal CA3 subfield. (**A**) Representative gels (inset) or changes in nuclear respiratory factor 1 (NRF1) relative to β-actin protein, detected in samples collected from the right CA3 subfield of the hippocampus in sham-control animals, or 6 h after the microinjection of KA (0.5 nmoL) into the left hippocampal CA3 subfield in animals received 24 h before pre-treatment with application into the bilateral CA3 subfield of antisense, sense, or scrambled (SC) ODNs against *Sirt1* (1 nmol). Antisense ODNs against *Sirt1* decreased seizure-induced NRF1 expression after 6 h following status epilepticus. (**B**) In accordance with NRF1 protein expression, the DNA-binding activity of NRF1 also decreased with *Sirt1* antisense ODNs. (**C**) Laser scanning confocal microscopic images of a representative cell in the right CA3b subfield of the hippocampus that was immunoreactive to a neuronal marker, NeuN (red fluorescence), and additionally stained for NRF1 (green fluorescence) in the sham-control animals (**C-a,b,c**), or 6 h after the microinjection of KA (0.5 nmol) into the left hippocampal CA3 subfield in animals received 24 h before pre-treatment with application into the bilateral CA3 subfield of the *Sirt1* antisense (**C-d,e,f**), sense (**C-g,h,i**), or SC (**C-j,k,l**) ODN (1 nmol). Scale bar, 10 μM. (**D**) Sirt1 antisense ODN lessened Tfam expression 6 h after KA treatment when compared with the sham animals, and the controls of sense and SC ODN for *Sirt1*. (**E**) Representative gels (inset) or changes in the content of mitochondrial protein COX1 relative to β-actin protein, detected in samples collected from the right CA3 subfield of the hippocampus in sham-control animals, or 24 h after the microinjection of KA (0.5 nmol) into the left hippocampal CA3 subfield in animals received 24 h before pre-treatment with application into the bilateral CA3 subfield of antisense, sense, or SC ODNs against *Sirt1* (1 nmol). (**F**) Quantification of mtDNA and presented with rat mtDNA to mouse mtDNA ratio in rats with the pre-treatment of *Sirt1* antisense, sense, or SC ODN, and sham animals, 24 h after induction of status epilepticus. M1: Mouse mtDNA control marker; M2: Rat mtDNA control marker. Values are the mean ± SEM of quadruplicate analyses from six animals per experimental group in (**A**,**D**–**F**). * *p* < 0.05 versus the sham control group, # *p* < 0.05 versus the *Sirt1* antisense ODN + KA group in the Scheffé multiple-range test.

**Figure 5 ijms-20-03588-f005:**
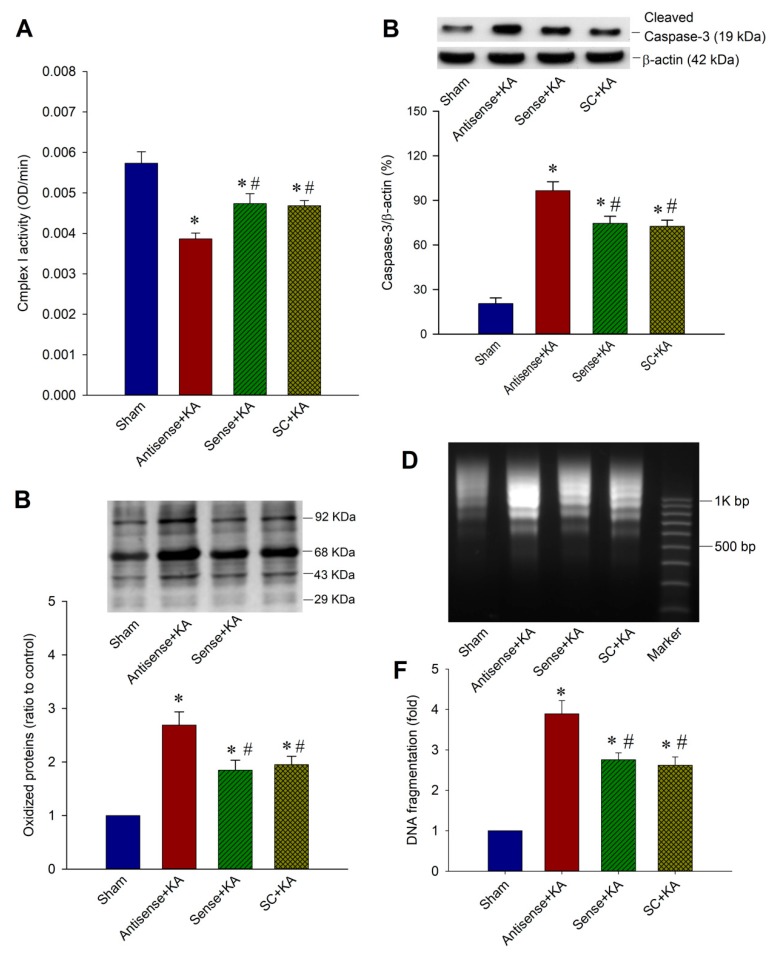
Pre-treatment of *Sirt1* antisense ODN augments related neuronal cell damage in the hippocampus following kainic acid (KA)-induced status epilepticus. (**A**) Enzyme assay for the activity of Complex I in the mitochondrial fraction isolated from the CA3 subfield of the hippocampus in the sham-control animals, or three days after microinjection of KA (0.5 nmol) into the left hippocampal CA3 subfield in animals which received a microinjection into the bilateral CA3 subfield of antisense ODN against *Sirt1*, or sense or scrambled (SC) ODN (1 nmol). (**B**) Representative gels (inset) or ratio to sham control change in protein oxidation detected in samples collected from the CA3 subfield of the hippocampus in animals that received *Sirt1* antisense, sense, or SC ODN. Total proteins were extracted from the hippocampal CA3 subfield in the sham-control animals, or seven days after the microinjection of KA (0.5 nmol) into the left hippocampal CA3 subfield in animals that received a microinjection into the bilateral CA3 subfield of antisense ODN against *Sirt1*, or sense or SC ODN (1 nmol), followed by immunoblot analysis for the extent of protein oxidation. (**C**) Representative gels (inset) or changes in activated caspase-3 (19 kDa) relative to β-actin protein, detected in the cytosolic fraction of samples collected from the CA3 subfield of the hippocampus in sham-control animals, or seven days after the microinjection of KA (0.5 nmol) into the left hippocampal CA3 subfield in animals that received a microinjection into the bilateral CA3 subfield of antisense ODN against *Sirt1*, or sense or SC ODN (1 nmol). (**D**) Qualitative or (**E**) quantitative analysis of DNA fragmentation detected in samples collected from the CA3 subfield of the hippocampus in sham-control animals, or seven days after the microinjection of KA (0.5 nmol) into the left hippocampal CA3 subfield in animals that received a microinjection into the bilateral CA3 subfield of antisense ODN against *Sirt1*, or sense or SC ODN (1 nmol). Values are the mean ± SEM of quadruplicate analyses from six animals per experimental group in (**A**–**C**,**F**). * *p* < 0.05 versus the sham control group, # *p* < 0.05 versus the antisense ODN + KA group in the Scheffé multiple-range test.

**Figure 6 ijms-20-03588-f006:**
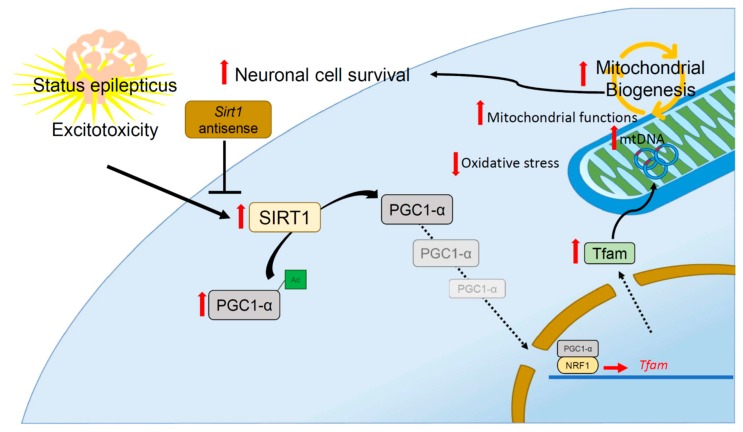
Schematic diagram illustrating the proposed SIRT1 signaling pathway involved in PGC1α/mitochondrial biogenesis that exerted an endogenous protective mechanism in the hippocampal neuronal tissue following experimental status epilepticus. Excitotoxicity induced by experimental status epilepticus may activate SIRT1 expression that can lead to the deacetylation (Ac) of PGC-1α by SIRT1 and induce mitochondrial biogenesis machinery expression in hippocampal neuronal cells. Downregulation of SIRT1 by *Sirt1* gene knockdown decreased mitochondrial biogenesis machinery expression, which was accompanied by impaired mitochondrial respiratory function, increased protein oxidation, and raised the apoptotic process and neuronal damage.

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
