# Peer review of "Sirtuin 1 Regulates Mitochondrial Biogenesis and Provides an Endogenous Neuroprotective Mechanism Against Seizure-Induced Neuronal Cell Death in the Hippocampus Following Status Epilepticus"

_ijms, 2019, doi:10.3390/ijms20143588_

Reviewer 1 Report

The study of SIRT1, as molecular target that modulates several essential molecular pathways involve in aging, plasticity, neurogenesis, inflammation, and protection against damage is without any doubt essential in the search of new treatments aim to prevent damage and promote cell survival, as it is the case of the status epilepticus. For this reason, this article presents relevant information. The topic is correctly addressed and enough clear. The methodology is well address and the bibliography is correctly updated. It is worth highlighting the quality of the figures.  However, some points should be clarified. 

Formal points:

1. Although at Figures it is cited, at Materials and Methods section is not cited the number of animals used in the experiment.

2. It could be useful a list of abbreviations.

Content points:

-As it is explain at the manuscript SIRT1 increased after injury as a protective mechanism, and as long as times goes by it decrease up to basal levels. It seems that it is an acutely activated system that acts as an alarm in cell protection. Therefore, how authors explain that levels of SIRT1 are decreased in other stress situations as aging or inflammation (as it is describe in bibliography)? This means that it is a system that has a different behavior depend on the temporality of the injury? or depending on the type of injury? Some explanation about this, should be mentioned at discussion.

-Authors study the relation between SIRT1, PGC-1a, complex I, oxidation, etc…but what about levels of cytokines and inflammation, or signaling pathways like Nf-kB, and antioxidant proteins like SOD1/2…Why authors delimitate the study in these proteins or systems? Which was the criterion?

- It is also describe that after injury for example high levels of ROS or cytokines mediates cleavage at the nuclear full-length SirT1 (FLSirT1; 110kDa) to generate a stable but enzymatically inactive 75kDa fragment of SirT1 (75SirT1) (Example: Deciphering the role of 75kDa SirT1 fragment in osteoarthritis. Dvir-Ginzberg, M. et al. Osteoarthritis and Cartilage, Volume 21, S49). Status epilepticus is a situation of acute injury authors know the effect of this status on the cleavage of SIRT1? , because it was glimpsed that part of the protection of SIRT1, a part from deacetylating some proteins as PGC-1a, it is a protective mechanism of SIRT1 fragments at mitochondrial level, which preserve cell survival, avoiding apoptosis.

-Which is the meaning of the results at functional levels? Accordingly to the results after 24 hours of the injury SIRT1, seems to be able to repair the system. This seems to be a too simple vision of the system and cell behavior, it is enough to return to the basal situation the induction of mitochondrial biogenesis machinery expression by SIRT1?

-The justification is well address, but in order to improve the quality of the article it could be useful to explain the perspective and future directions, which are the specific future questions to be answer? What kind of information is needed, in which models, in order to could use this information, for finding specific translational solutions? Authors can explain this point better.

Author Response

Responses to Reviewer #1

Formal points:

1.      Although at Figures it is cited, at Materials and Methods section is not cited the number of animals used in the experiment.

Response: “Experiments were carried out in 360 specific pathogen-free adult male Sprague-Dawley rats”had been cited in Materials and Methods section (Line 417).

2.      It could be useful a list of abbreviations.

Response: the list of abbreviations in Line 575.

Content points:

1.      -As it is explain at the manuscript SIRT1 increased after injury as a protective mechanism, and as long as times goes by it decrease up to basal levels. It seems that it is an acutely activated system that acts as an alarm in cell protection. Therefore, how authors explain that levels of SIRT1 are decreased in other stress situations as aging or inflammation (as it is describe in bibliography)? This means that it is a system that has a different behavior depend on the temporality of the injury? or depending on the type of injury? Some explanation about this, should be mentioned at discussion.

Response: Whereas the levels of SIRT1 may decrease in situations such as aging or inflammation [1,2], evidence demonstrated the SIRT1 signaling pathway was activated in many neurological diseases [3-12]. This diverse results may be related to the different physiological conditions or pathological injuries. We have enabled the discussion in Lines 349-353.

2. -Authors study the relation between SIRT1, PGC-1a, complex I, oxidation, etc…but what about levels of cytokines and inflammation, or signaling pathways like Nf-kB, and antioxidant proteins like SOD1/2…Why authors delimitate the study in these proteins or systems? Which was the criterion?

Response: thank you for your suggestion, the role of neuroinflammation and antioxidation in PGC-1a had been well studied in our previous works (please see our previous literatures). However, we will explore the SIRT1 in the role of neuroinflammation in the future works.

3. - It is also describe that after injury for example high levels of ROS or cytokines mediates cleavage at the nuclear full-length SirT1 (FLSirT1; 110kDa) to generate a stable but enzymatically inactive 75kDa fragment of SirT1 (75SirT1) (Example: Deciphering the role of 75kDa SirT1 fragment in osteoarthritis. Dvir-Ginzberg, M. et al. Osteoarthritis and Cartilage, Volume 21, S49). Status epilepticus is a situation of acute injury authors know the effect of this status on the cleavage of SIRT1? , because it was glimpsed that part of the protection of SIRT1, a part from deacetylating some proteins as PGC-1a, it is a protective mechanism of SIRT1 fragments at mitochondrial level, which preserve cell survival, avoiding apoptosis. Response: Thank you for your suggestion. You let us to understand the role of 75kDa fragment of SirT1 (75SirT1). We have added the notion and discussion in the “discussion section” (Lines: 209-302).

4. -Which is the meaning of the results at functional levels? Accordingly to the results after 24 hours of the injury SIRT1, seems to be able to repair the system. This seems to be a too simple vision of the system and cell behavior, it is enough to return to the basal situation the induction of mitochondrial biogenesis machinery expression by SIRT1?

Response: While we could not find a significant change in the electrophysiology during the control SIRT1 pathway, however, the neuronal damage significantly decreased in this study, whether the functional alternation may require the further studies.

5. -The justification is well address, but in order to improve the quality of the article it could be useful to explain the perspective and future directions, which are the specific future questions to be answer? What kind of information is needed, in which models, in order to could use this information, for finding specific translational solutions? Authors can explain this point better.

Response: We are very appreciate your suggestion. We re-wrote the conclusion, to emphasis the future directions (Line 565-574).

 We thus appreciate very much the opportunity to improve on our manuscript; and sincerely hope that our revision will now meet with your approval for publication in International Journal of Molecular Sciences.

Respectfully submitted,

Yao-Chung Chuang, MD, PhD.

Professor of Neurology; Kaohsiung Chang Gung Memorial Hospital, Chang Gung University, Taiwan

1.         Ng, F.; Wijaya, L.; Tang, B.L. SIRT1 in the brain-connections with aging-associated disorders and lifespan. Front Cell Neurosci 2015, 9, 64, doi:10.3389/fncel.2015.00064.

2.         Chang, H.C.; Guarente, L. SIRT1 mediates central circadian control in the SCN by a mechanism that decays with aging. Cell 2013, 153, 1448-1460, doi:10.1016/j.cell.2013.05.027.

3.         Zhang, F.; Wang, S.; Gan, L.; Vosler, P.S.; Gao, Y.; Zigmond, M.J.; Chen, J. Protective effects and mechanisms of sirtuins in the nervous system. Prog Neurobiol 2011, 95, 373-395, doi:10.1016/j.pneurobio.2011.09.001.

4.         Xu, J.; Jackson, C.W.; Khoury, N.; Escobar, I.; Perez-Pinzon, M.A. Brain SIRT1 Mediates Metabolic Homeostasis and Neuroprotection. Front Endocrinol (Lausanne) 2018, 9, 702, doi:10.3389/fendo.2018.00702.

5.         Rizzi, L.; Roriz-Cruz, M. Sirtuin 1 and Alzheimer's disease: An up-to-date review. Neuropeptides 2018, 71, 54-60, doi:10.1016/j.npep.2018.07.001.

6.         Singh, P.; Hanson, P.S.; Morris, C.M. SIRT1 ameliorates oxidative stress induced neural cell death and is down-regulated in Parkinson's disease. BMC Neurosci 2017, 18, 46, doi:10.1186/s12868-017-0364-1.

7.         Wan, D.; Zhou, Y.; Wang, K.; Hou, Y.; Hou, R.; Ye, X. Resveratrol provides neuroprotection by inhibiting phosphodiesterases and regulating the cAMP/AMPK/SIRT1 pathway after stroke in rats. Brain Res Bull 2016, 121, 255-262, doi:10.1016/j.brainresbull.2016.02.011.

8.         Zhang, J.F.; Zhang, Y.L.; Wu, Y.C. The Role of Sirt1 in Ischemic Stroke: Pathogenesis and Therapeutic Strategies. Front Neurosci 2018, 12, 833, doi:10.3389/fnins.2018.00833.

9.         Folbergrova, J.; Jesina, P.; Kubova, H.; Otahal, J. Effect of Resveratrol on Oxidative Stress and Mitochondrial Dysfunction in Immature Brain during Epileptogenesis. Mol Neurobiol 2018, 55, 7512-7522, doi:10.1007/s12035-018-0924-0.

10.       Wu, Z.; Xu, Q.; Zhang, L.; Kong, D.; Ma, R.; Wang, L. Protective effect of resveratrol against kainate-induced temporal lobe epilepsy in rats. Neurochem Res 2009, 34, 1393-1400, doi:10.1007/s11064-009-9920-0.

11.       Castro, O.W.; Upadhya, D.; Kodali, M.; Shetty, A.K. Resveratrol for Easing Status Epilepticus Induced Brain Injury, Inflammation, Epileptogenesis, and Cognitive and Memory Dysfunction-Are We There Yet? Front Neurol 2017, 8, 603, doi:10.3389/fneur.2017.00603.

12.       Li, Z.; You, Z.; Li, M.; Pang, L.; Cheng, J.; Wang, L. Protective Effect of Resveratrol on the Brain in a Rat Model of Epilepsy. Neurosci Bull 2017, 33, 273-280, doi:10.1007/s12264-017-0097-2.

Responses to Reviewer #1

Formal points:

1.         Although at Figures it is cited, at Materials and Methods section is not cited the number of animals used in the experiment.

Response: “Experiments were carried out in 360 specific pathogen-free adult male Sprague-Dawley rats”had been cited in Materials and Methods section (Line 417).

2.         It could be useful a list of abbreviations.

Response: the list of abbreviations in Line 575.

Content points:

1.         -As it is explain at the manuscript SIRT1 increased after injury as a protective mechanism, and as long as times goes by it decrease up to basal levels. It seems that it is an acutely activated system that acts as an alarm in cell protection. Therefore, how authors explain that levels of SIRT1 are decreased in other stress situations as aging or inflammation (as it is describe in bibliography)? This means that it is a system that has a different behavior depend on the temporality of the injury? or depending on the type of injury? Some explanation about this, should be mentioned at discussion.

Response: Whereas the levels of SIRT1 may decrease in situations such as aging or inflammation [1,2], evidence demonstrated the SIRT1 signaling pathway was activated in many neurological diseases [3-12]. This diverse results may be related to the different physiological conditions or pathological injuries. We have enabled the discussion in Lines 349-353.

2. -Authors study the relation between SIRT1, PGC-1a, complex I, oxidation, etc…but what about levels of cytokines and inflammation, or signaling pathways like Nf-kB, and antioxidant proteins like SOD1/2…Why authors delimitate the study in these proteins or systems? Which was the criterion?

Response: thank you for your suggestion, the role of neuroinflammation and antioxidation in PGC-1a had been well studied in our previous works (please see our previous literatures). However, we will explore the SIRT1 in the role of neuroinflammation in the future works.

3. - It is also describe that after injury for example high levels of ROS or cytokines mediates cleavage at the nuclear full-length SirT1 (FLSirT1; 110kDa) to generate a stable but enzymatically inactive 75kDa fragment of SirT1 (75SirT1) (Example: Deciphering the role of 75kDa SirT1 fragment in osteoarthritis. Dvir-Ginzberg, M. et al. Osteoarthritis and Cartilage, Volume 21, S49). Status epilepticus is a situation of acute injury authors know the effect of this status on the cleavage of SIRT1? , because it was glimpsed that part of the protection of SIRT1, a part from deacetylating some proteins as PGC-1a, it is a protective mechanism of SIRT1 fragments at mitochondrial level, which preserve cell survival, avoiding apoptosis. Response: Thank you for your suggestion. You let us to understand the role of 75kDa fragment of SirT1 (75SirT1). We have added the notion and discussion in the “discussion section” (Lines: 209-302).

4. -Which is the meaning of the results at functional levels? Accordingly to the results after 24 hours of the injury SIRT1, seems to be able to repair the system. This seems to be a too simple vision of the system and cell behavior, it is enough to return to the basal situation the induction of mitochondrial biogenesis machinery expression by SIRT1?

Response: While we could not find a significant change in the electrophysiology during the control SIRT1 pathway, however, the neuronal damage significantly decreased in this study, whether the functional alternation may require the further studies.

5. -The justification is well address, but in order to improve the quality of the article it could be useful to explain the perspective and future directions, which are the specific future questions to be answer? What kind of information is needed, in which models, in order to could use this information, for finding specific translational solutions? Authors can explain this point better.

Response: We are very appreciate your suggestion. We re-wrote the conclusion, to emphasis the future directions (Line 565-574).

Reviewer 2 Report

Role of Sirtuin 1 in an animal model of status epilepticus is well-described in this manuscript. All the experiments seem technically sound and their results are of interest.

However, authors will have to brush up their English writing to make this paper more persuasive. There are too many typos and grammatical errors. Also, please try to make the discussion section more concise by cutting redundant explanations about previous literatures.

Author Response

Response to Reviewer #2

1.      Role of Sirtuin 1 in an animal model of status epilepticus is well-described in this manuscript. All the experiments seem technically sound and their results are of interest.

Response: Thank you very much and your nice comment.

2.      However, authors will have to brush up their English writing to make this paper more persuasive. There are too many typos and grammatical errors. Also, please try to make the discussion section more concise by cutting redundant explanations about previous literatures.

Response: We have sent my manuscript to English editing (MDPI's English editing service. 4d397d958af95f60), Also we have elaborated our dissection.

 We thus appreciate very much the opportunity to improve on our manuscript; and sincerely hope that our revision will now meet with your approval for publication in International Journal of Molecular Sciences.

Respectfully submitted,

Yao-Chung Chuang, MD, PhD.

Professor of Neurology; Kaohsiung Chang Gung Memorial Hospital, Chang Gung University, Taiwan

Response to Reviewer #2

1.      Role of Sirtuin 1 in an animal model of status epilepticus is well-described in this manuscript. All the experiments seem technically sound and their results are of interest.

Response: Thank you very much and your nice comment.

2.      However, authors will have to brush up their English writing to make this paper more persuasive. There are too many typos and grammatical errors. Also, please try to make the discussion section more concise by cutting redundant explanations about previous literatures.

Response: We have sent my manuscript to English editing (MDPI's English editing service. 4d397d958af95f60), Also we have elaborated our dissection.

 We thus appreciate very much the opportunity to improve on our manuscript; and sincerely hope that our revision will now meet with your approval for publication in International Journal of Molecular Sciences.

Respectfully submitted,

Yao-Chung  Chuang, MD, PhD.

Professor of Neurology; Kaohsiung Chang Gung Memorial Hospital, Chang Gung University, Taiwan

1.         Ng, F.; Wijaya, L.; Tang, B.L. SIRT1 in the brain-connections with aging-associated disorders and lifespan. Front Cell Neurosci 2015, 9, 64, doi:10.3389/fncel.2015.00064.

2.         Chang, H.C.; Guarente, L. SIRT1 mediates central circadian control in the SCN by a mechanism that decays with aging. Cell 2013, 153, 1448-1460, doi:10.1016/j.cell.2013.05.027.

3.         Zhang, F.; Wang, S.; Gan, L.; Vosler, P.S.; Gao, Y.; Zigmond, M.J.; Chen, J. Protective effects and mechanisms of sirtuins in the nervous system. Prog Neurobiol 2011, 95, 373-395, doi:10.1016/j.pneurobio.2011.09.001.

4.         Xu, J.; Jackson, C.W.; Khoury, N.; Escobar, I.; Perez-Pinzon, M.A. Brain SIRT1 Mediates Metabolic Homeostasis and Neuroprotection. Front Endocrinol (Lausanne) 2018, 9, 702, doi:10.3389/fendo.2018.00702.

5.         Rizzi, L.; Roriz-Cruz, M. Sirtuin 1 and Alzheimer's disease: An up-to-date review. Neuropeptides 2018, 71, 54-60, doi:10.1016/j.npep.2018.07.001.

6.         Singh, P.; Hanson, P.S.; Morris, C.M. SIRT1 ameliorates oxidative stress induced neural cell death and is down-regulated in Parkinson's disease. BMC Neurosci 2017, 18, 46, doi:10.1186/s12868-017-0364-1.

7.         Wan, D.; Zhou, Y.; Wang, K.; Hou, Y.; Hou, R.; Ye, X. Resveratrol provides neuroprotection by inhibiting phosphodiesterases and regulating the cAMP/AMPK/SIRT1 pathway after stroke in rats. Brain Res Bull 2016, 121, 255-262, doi:10.1016/j.brainresbull.2016.02.011.

8.         Zhang, J.F.; Zhang, Y.L.; Wu, Y.C. The Role of Sirt1 in Ischemic Stroke: Pathogenesis and Therapeutic Strategies. Front Neurosci 2018, 12, 833, doi:10.3389/fnins.2018.00833.

9.         Folbergrova, J.; Jesina, P.; Kubova, H.; Otahal, J. Effect of Resveratrol on Oxidative Stress and Mitochondrial Dysfunction in Immature Brain during Epileptogenesis. Mol Neurobiol 2018, 55, 7512-7522, doi:10.1007/s12035-018-0924-0.

10.       Wu, Z.; Xu, Q.; Zhang, L.; Kong, D.; Ma, R.; Wang, L. Protective effect of resveratrol against kainate-induced temporal lobe epilepsy in rats. Neurochem Res 2009, 34, 1393-1400, doi:10.1007/s11064-009-9920-0.

11.       Castro, O.W.; Upadhya, D.; Kodali, M.; Shetty, A.K. Resveratrol for Easing Status Epilepticus Induced Brain Injury, Inflammation, Epileptogenesis, and Cognitive and Memory Dysfunction-Are We There Yet? Front Neurol 2017, 8, 603, doi:10.3389/fneur.2017.00603.

12.       Li, Z.; You, Z.; Li, M.; Pang, L.; Cheng, J.; Wang, L. Protective Effect of Resveratrol on the Brain in a Rat Model of Epilepsy. Neurosci Bull 2017, 33, 273-280, doi:10.1007/s12264-017-0097-2.